# Light-switchable transcription factors obtained by direct screening in mammalian cells

Liyuan Zhu[1], Harold M. McNamara[2,3] & Jared E. Toettcher [2]✉

Optogenetic tools can provide fine spatial and temporal control over many biological processes. Yet the development of new light-switchable protein variants remains challenging, and the field still lacks general approaches to engineering or discovering protein variants with light-switchable biological functions. Here, we adapt strategies for protein domain insertion and mammalian-cell expression to generate and screen a library of candidate optogenetic tools directly in mammalian cells. The approach is based on insertion of the AsLOV2 photoswitchable domain at all possible positions in a candidate protein of interest, introduction of the library into mammalian cells, and light/dark selection for variants with photoswitchable activity. We demonstrate the approach's utility using the Gal4-VP64 transcription factor as a model system. Our resulting LightsOut transcription factor exhibits a > 150-fold change in transcriptional activity between dark and blue light conditions. We show that light-switchable function generalizes to analogous insertion sites in two additional Cys6Zn2 and C2H2 zinc finger domains, providing a starting point for optogenetic regulation of a broad class of transcription factors. Our approach can streamline the identification of single-protein optogenetic switches, particularly in cases where structural or biochemical knowledge is limited.

Optogenetic tools are increasingly used to interrogate a wide range of biological processes, joining small molecule inhibitors and agonists as standard perturbative tools available to cell and developmental biologists[1]. Yet despite their widespread adoption, light-switchable variants are still only available for a limited subset of highly studied proteins and pathways. One limitation faced by the field is the development of general strategies to obtain light-switchable variants of any protein of interest. Without detailed knowledge of a target protein's structure and mechanism of activation, it remains challenging to incorporate photoswitchable domains in a manner that enables light-gated regulation of the target protein's activity.

One promising technique for engineering photoswitchable protein variants involves the internal insertion of a light-sensitive domain such as the LOV2 domain from *Avena sativa* Phototropin 1 (AsLOV2). Upon illumination, a conformational change in the light-sensitive domain can then propagate to the attached protein, potentially altering its functional state. Indeed, studies found that the AsLOV2 domain could be used to allosterically control proteins of interest, provided that a suitable insertion site could be found[2–4]. Since these initial studies, similar approaches have been used to develop light-controlled nanobodies and monobodies against multiple target proteins[5,6], light- and temperature-controlled systems for regulating Cas9[7,8], and light-based regulation of mammalian pyruvate kinase activity[9].

While powerful, the LOV-insertion strategy is still limited by the difficulty of finding a suitable insertion site to mediate a strong photoswitchable effect. One challenge is that the effect of AsLOV2

[1]Department of Chemistry, Princeton University, Princeton, NJ 08544, USA. [2]Department of Molecular Biology, Princeton University, Princeton, NJ 08544, USA. [3]Lewis Sigler Institute, Princeton University, Princeton, NJ 08544, USA. ✉e-mail: toettcher@princeton.edu

insertion can vary substantially in response to even subtle changes in insertion site or linker length. For example, in our own prior study to design photoswitchable nanobodies, we found a high degree of variability in light-dependent binding even from insertion sites within the same surface-exposed loop[5]. Even varying the insertion site or linker lengths by 1-2 residues could substantially alter or abolish photoswitchable activity. Another challenge is that LOV domain insertion might alter protein activity in myriad ways, including direct allosteric switching of the fusion protein's function[2–4], occluding an interaction surface with other host proteins[8,10], or eliciting changes in its subcellular localization or expression in cells[5]. Many of these effects may not be well-captured in structural analyses of the isolated protein and may not translate from mammalian cells to bacterial systems or in vitro assays, suggesting that it may be especially powerful to directly evaluate candidates using functional assays in a target cell type of interest.

How might the generality of the LOV-insertion approach be improved for discovery of mammalian-cell optogenetic tools? We reasoned that an ideal approach would (1) test all possible AsLOV2 insertion sites in a target protein, thus eliminating a requirement for prior knowledge about the target's structure or function and overcoming the changes in activity that can result from small shifts in insertion site, and (2) would assess light-dependent activity directly in mammalian cells, thereby enabling discovery of complex photoswitchable functions (e.g., metazoan genome regulation) that may not be easily transferable to bacteria, yeast, or in vitro assays. Despite the utility of a direct mammalian-cell optogenetic screen, such an approach faces numerous challenges including generating a high-coverage library of protein insertions, introducing a single library member per cell at comparable expression levels, and performing sequences of selection under light and dark conditions to identify photoswitchable variants.

Here, we set out to overcome these challenges and generalize the LOV-insertion technique to screen a library of candidate photoswitchable proteins directly in mammalian cells (Fig. 1a). Our strategy relies on the Mu transposase to randomly insert the LOV sequence into a target protein of interest[11], followed by genomic integration of individual library members into a mammalian cell line using the recently developed landing pad system[12,13]. Successive selection experiments under light and dark conditions can then be used to identify library members with photoswitchable response functions. Applying this screening approach to a model target protein, the Gal4-VP64 transcription factor, enabled us to obtain a light-sensitive variant that exhibited a 150-fold light-dependent shift in transcriptional activity. Mechanistic studies of the optimal AsLOV2 insertion variant demonstrated that the large light-induced change in gene expression depended on tight conformational coupling between the AsLOV2 and Gal4 domains, but was not correlated with large changes in protein levels, localization, or in vitro DNA binding affinity. We further showed that photoswitchable function could be extended to other zinc finger DNA binding domains, suggesting that the insertion site found in our screen may generalize to a broad family of DNA-binding proteins. Overall, our results suggest that high-throughput identification of optogenetic tools controlling complex biological responses may be within reach, even for targets with poorly characterized structure or domain organization.

## Results
### Establishing AsLOV2 insertion libraries into a gene of interest
Our overall goal was to perform a screen for candidate light-switchable transcription factors directly in mammalian cells. We first set out to construct a library of Gal4-VP64 transcription factor variants with the photoswitchable AsLOV2 domain inserted at different amino acid positions. We used a recent technique termed domain insertion profiling with DNA sequencing (DIP-Seq), originally developed for screening sites of fluorescent protein insertion[11]. This technique relies on insertion of a chloramphenicol resistance (CmR) cassette into an ampicillin-resistant plasmid of interest using the Mu transposase[14,15], followed by combined chloramphenicol/ampicillin selection to obtain a library of insertion variants. Subsequent digestion using a Type IIS restriction enzyme (BsaI) removes the resistance cassette and part of the transposon scar, enabling insertion of a sequence of interest at the transposition site into each linearized plasmid. Digestion with a second Type IIS enzyme (BsmBI) is subsequently used to move the fusion protein library into an expression vector for introduction into cells and subsequent analysis.

We applied the DIP-Seq protocol to generate CmR cassette insertions into plasmid pLZA066, a pUC19 target plasmid containing residues 1-147 of Gal4 fused to the VP64 transcription activation domain (Fig. 1b). Quantification of the number of resistant colonies revealed approximately 190-fold coverage of possible insertion sites in the pUC19 Gal4-VP64 plasmid (Supplementary Table 1), consistent with the high coverage obtained for Mu transposition in prior studies for even large proteins of >1000 amino acids (e.g., Cas9)[16]. After drug selection for transposition, we replaced the CmR selection cassette with the AsLOV2 coding sequence by BsaI digestion and ligation (Fig. 1c). We used a slightly shorter AsLOV2 sequence (residues 408–543 of *Avena sativa* Phototropin (1) compared to the canonical sequence (residues 404–546), based on our prior study indicating that this shortened sequence showed improved photoswitching when inserted into a target protein[5,6]. Reasoning that our AsLOV2 sequence might be inserted in any codon reading frame, we prepared +0, +1, and +2 frameshifted versions of the AsLOV2 library (libraries pLZA066_LOV1, pLZA066_LOV2, and pLZA066_LOV3, respectively) to ensure the possibility of in-frame insertion at every nucleotide position (see Methods). We then performed BsmBI digestion and gel purification to specifically capture Gal4-VP64 constructs containing the AsLOV2 insertion (Fig. 1d). The library of AsLOV2-inserted Gal4-VP64 sequences were then cloned into the pLZA063 landing pad integration plasmid (discussed in detail below), resulting in +0, +1 and +2-shifted AsLOV2 insertion libraries pLZA063-LOV01, pLZA063-LOV02 and pLZA063-LOV03, respectively. We performed next-generation sequencing to map the frequency of AsLOV2 insertions in the pLZA063-LOV02 library (Fig. 1e); as expected, we observed high coverage of AsLOV2 insertion sites specifically within the Gal4-VP64 coding sequence. Overall, these results support DIP-Seq as an excellent approach for generating insertion libraries of candidate optogenetic tools.

### Expression of the AsLOV2 insertion library in mammalian cells
We next set out to establish a mammalian cell line for hosting and screening the photoswitchable Gal4-VP64 library (Fig. 2a). We took advantage of a recently established technology, the landing pad system[12,13], that enables the insertion of a promoter-less target gene cassette into a single defined genomic locus that harbors a doxycycline-inducible promoter to drive expression only from the successfully-integrated cassette. In this scheme, bulk transfection with the entire AsLOV2 insertion library would be expected to result in a single stably integrated and expressed variant per cell. Integration into the landing pad further leads to displacement of a BFP gene, and we linked Gal4-VP64 expression to mCherry expression using an internal ribosomal entry sequence (IRES) so that cells with successful landing pad integration would be expected to be BFP-/mCherry+ (Fig. 2b).

We adopted a HEK293T landing pad (293-LP) cell line as our screening platform due to their high transfection efficiency and ease of growth. Initial experiments confirmed that transfecting 293-LP cells with a mixed population of GFP and mCherry landing pad integration cassettes in the presence of 2 µg/mL doxycycline resulted in either GFP+ or mCherry+ cells, but not both, indicative of gene expression driven from a single integrant per cell at the landing pad site (Supplementary Fig. 1a). We further generated a clonal 293-LP cell line with a

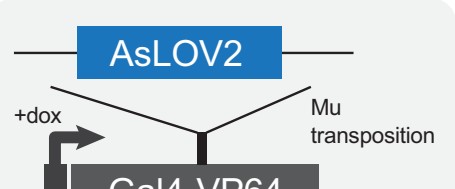

**a** 1. Construct library of candidate photoswitchable Gal4 TFs

2. Generate mammalian cell insertion library

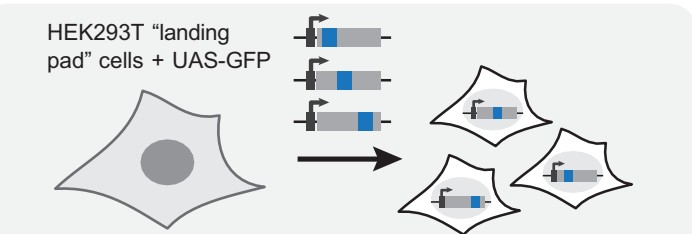

3. Select for photoswitchable function

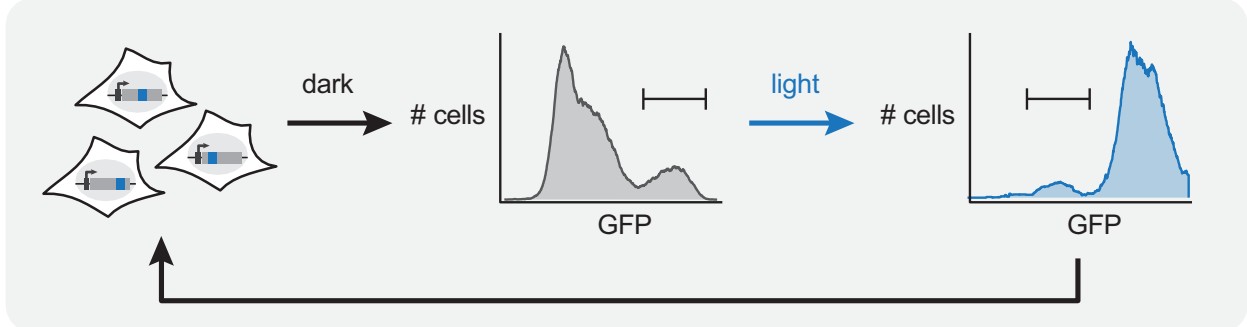

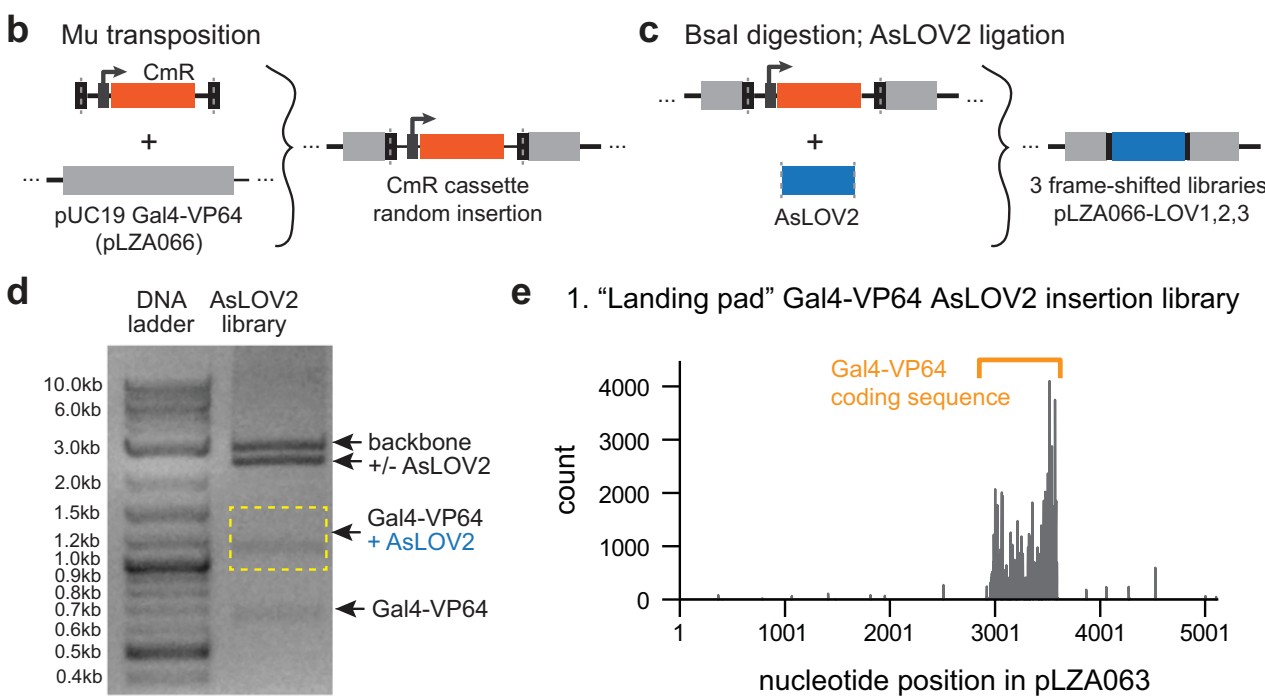

**b** Mu transposition

**c** BsaI digestion; AsLOV2 ligation

**d**

**e** 1. "Landing pad" Gal4-VP64 AsLOV2 insertion library

**Fig. 1 | Screening a library of candidate optogenetic tools in mammalian cells.** **a** Schematic of overall approach. To screen for insertion sites, we first randomly insert AsLOV2 into a target protein of interest (here, Gal4-VP64) using Mu transposition. A mammalian cell line harboring a single landing pad site and a GFP-based reporter of target protein activity is used as a chassis for the screen. The AsLOV2-inserted library is incorporated as a single variant per cell. GFP selection under dark and light conditions identifies variants with photo-switchable function. **b** Schematic of initial transposon insertion step, where a chloramphenicol resistance cassette is inserted into a pUC19 plasmid containing the target gene. BsaI restriction sites within the Mu recognition sequence are shown (dotted gray lines).

**c** In a second step, BsaI digestion and ligation inserts AsLOV2 and eliminates a portion of the Mu recognition sequence to minimize linker length. **d** A second restriction digestion is used to purify AsLOV2-inserted target sequence for subsequent cloning into mammalian expression vector. The same experiment has been repeated at least five times with similar results. **e** Next-generation sequencing of the final mammalian expression vector identifies AsLOV2 insertions specifically within the Gal4-VP64 target gene. Raw sequencing data was available in Sequence Read Archive (SRA) with the accession code SAMN35215817. Source data for (**d** and **e**) are provided as a Source Data file.

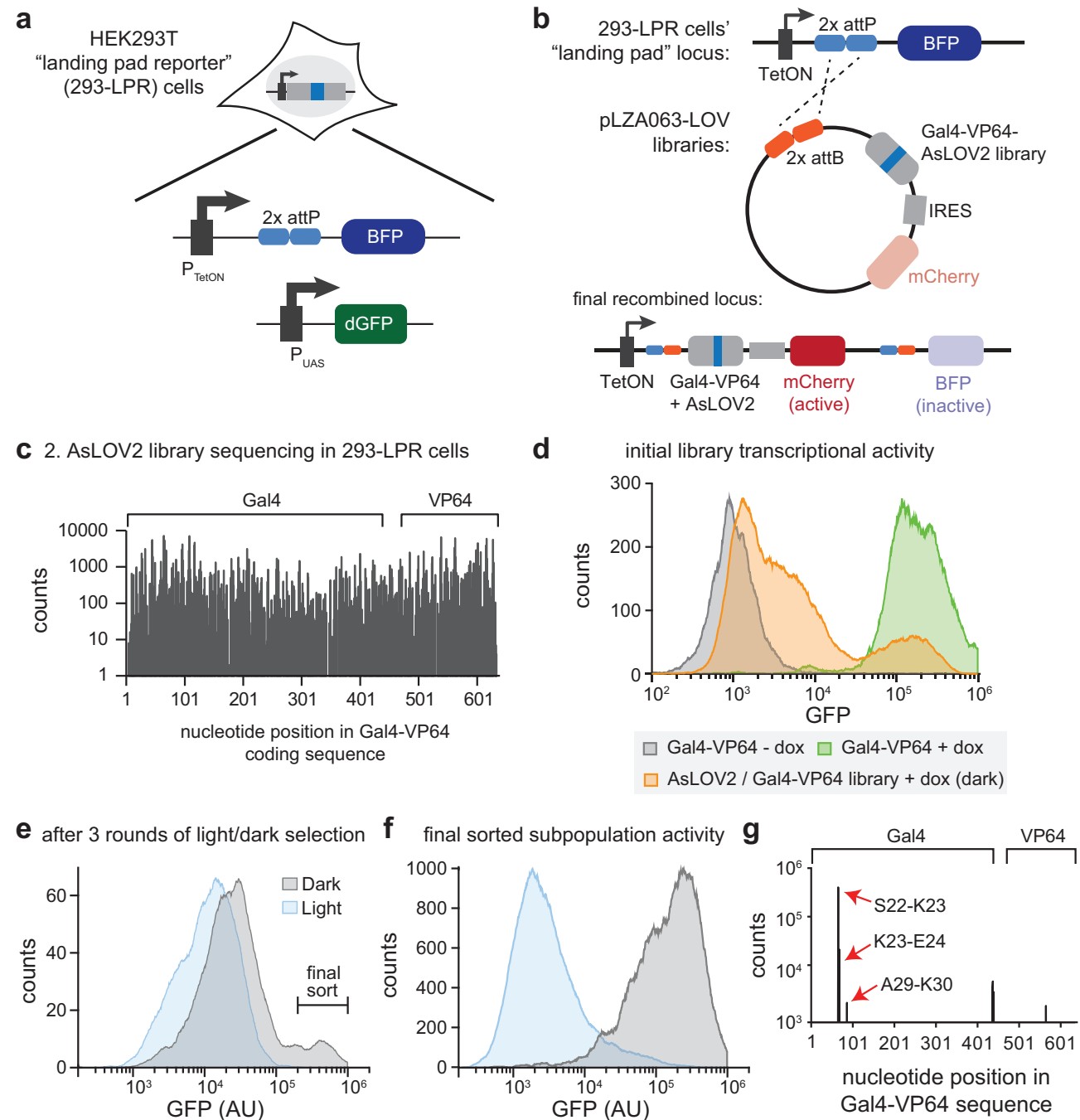

**Fig. 2 | Selecting for photoswitchable transcription in mammalian cells.**
**a**, **b** Genomic composition of landing pad reporter system, showing the initial landing pad locus (**a**) and its modification upon incorporation of a library variant (**b**). A single library member can be inserted into the landing pad locus, driving expression from a doxycycline-inducible promoter (TetON) and displacing the BFP gene. After integration, expression of the library variant is also linked to mCherry expression. GFP expression is then dependent on the activity of individual library variant in each cell. **c** Sequencing of AsLOV2 insertion in 293-LPR cells harboring the final pLZA063-LOV02 library indicates coverage throughout the Gal4-VP64 coding sequence, with >80% of insertions observed at least once. Counts are displaced as log(x) + 1 to accommodate the positions with zero counts in a logarithm-scale plot. Raw sequencing data was available in SRA with the accession code SAMN35215818. **d** Single-cell GFP distribution of initial library, with most cells lacking GFP

expression, a few cells retaining GFP level comparable to that of wild type Gal4 and some cells displaying intermediate GFP level. **e** After three rounds of dark/light selection, a GFP-positive subpopulation in the dark that was not present under blue light illumination was observed and then sorted. See Supplementary Figs. 2 and 9 for flow cytometry gating procedures. **f** Library after the last round of sorting exhibited a strong light-dependent transcriptional response. **g** AsLOV2 insertion distribution for the library after last sorting. A single cluster of insertion sites in Gal4 was observed, with the insertion between Ser 22 and Lys 23 as the predominant hit. Counts are displaced as log(x) + 1 to accommodate the positions with zero counts in a logarithm-scale plot. Raw sequencing data was available in SRA with the accession code SAMN35215819. Source data for (**c**–**g**) are provided as a Source Data file. AU arbitrary units.

stably-integrated, Gal4-responsive UAS promoter driving expression of destabilized GFP (P_UAS-GFP) to assess the transcriptional activity of each Gal4-VP64 AsLOV2 variant. We observed robust GFP expression after transfection of a Gal4-VP64 expression construct and culture in doxycycline-containing media (Supplementary Fig. 1b). We termed the resulting clonal cell line HEK293T landing pad reporter cells (293-LPR) cells.

We next transfected the three frame-shifted libraries into the 293-LPR cell line and sorted 100,000 mCherry+/BFP− cells in each case to form library-expressing cell lines for subsequent screening (Supplementary Fig. 1c). We measured the diversity of the pLZA063-LOV02 library after stable landing pad integration in 293-LPR cells using DNA sequencing (Fig. 2c), which revealed that 80.3% of possible AsLOV2 insertion sites into Gal4-VP64 were observed at least once, a result comparing favorably to prior DIP-seq results[11]. Assuming similar coverage in all three libraries, this result would suggest that >99% of amino acid insertion sites are represented in at least one of the three libraries. We next measured single-cell GFP expression in cells expressing the pLZA063-LOV libraries to assess Gal4 activity (Fig. 2d; Supplementary Fig. 1d). Many cells lacked GFP expression, indicating a complete loss of Gal4 activity that would be expected from out-of-frame or non-functional insertions of AsLOV2 into the Gal4-VP64 coding sequence. A small subpopulation expressed GFP at high levels similar to unmodified Gal4-VP64, suggesting that some AsLOV2 insertions had no deleterious effect. Finally, we observed some cells with intermediate GFP levels, suggesting partial perturbation of function. Taken together, these data that AsLOV2 insertion drives diverse transcriptional responses in a mammalian cell library, providing a rich starting point for subsequent screening.

## Screening for photoswitchable gene expression in mammalian cells

To select for photoswitchable Gal4-VP64 variants, we performed a sequence of screening experiments in which cells were incubated in darkness or blue light for 22 h and sorted into GFP-high or GFP-low subpopulations, respectively. We initially recovered the GFP-positive population from our initial library in the dark in order to eliminate AsLOV2 domain insertions that destroyed Gal4-VP64 function (Supplementary Fig. 2a). After expanding the recovered population, we incubated it in the light and sorted the GFP-low population, repeating this procedure for subsequent selection rounds (Supplementary Fig. 2b). Cell sorting gates were initially chosen to be broad, capturing a wide range of GFP levels, and progressively refined in subsequent rounds of selection. At each selection stage we also measured the entire cell population's GFP expression in both light and dark conditions to detect any light-dependent shift in Gal4-VP64 activity (Supplementary Fig. 2c). After three rounds of light/dark selection of cells accommodating the pLZA063-LOV02 library we observed a prominent light-dependent transcriptional response: a GFP-positive subpopulation in the dark that was not present under blue light illumination (Fig. 2e). Sorting this population and measuring its light-dependent response revealed strong suppression of GFP expression under blue light, with a dark response comparable to unmodified Gal4 (Fig. 2f).

We next sought to determine which AsLOV2 insertion sites were responsible for the photoswitchable effect. Sequencing the pLZA063-LOV02 library at each stage of selection revealed the dynamics with which specific AsLOV2 insertions were enriched. For example, we found that the initial GFP-positive population included many insertions in the unstructured VP64 transcriptional activation domain, including some frameshift mutations that prevented translation of the final VP64 C-terminal amino acids (Supplementary Fig. 2d). This result can be rationalized by noting that VP64 itself consists of four repeats of a VP16 transcriptional activation domain, any one of which is sufficient to drive some GFP expression. As a result, any insertion that destroyed only a few VP16 repeats would be expected to retain at least some GFP

expression, albeit without any photoswitchable response. These mutations were excluded by subsequent rounds of sorting as they failed to exhibit low GFP under blue light, and the final light-switchable population predominantly consisted of single cluster of insertions within the zinc finger DNA binding domain. Within this cluster AsLOV2 was inserted after either Ser22, Lys23, or Ala29, with insertion after Ser22 identified as the most prevalent hit (Fig. 2g).

## Characterization of the LightsOut Gal4-VP64 transcription factor

We next set out to characterize light-induced transcription changes for the hits identified in our screen. We transiently transfected 293-LPR cells with plasmids that constitutively expressed Gal4-VP64, Gal4LOV^SK22-VP64, Gal4LOV^KE23-VP64, or Gal4LOV^AK29-VP64, and compared gene expression in light and dark (Fig. 3a). We observed light-dependent GFP expression from each AsLOV2 insertion variant, with the strength of each variant corresponding to the order of their appearance in the screen. The most potent variant, with AsLOV2 inserted after position Ser22 (Gal4LOV^SK22-VP64), exhibited a 156-fold change in gene expression between light and dark (Fig. 3a), comparable to other state-of-the-art optogenetic transcription systems[17,18]. These data also confirm that light-induced transcription was not doxycycline-dependent, an important control due to prior reports of light-dependent doxycycline degradation[19]. We termed our best-performing variant (Gal4LOV^SK22-VP64) the LightsOut system, following the convention of another optogenetic Gal4 variant, the LightOn system[18].

LightsOut-expressing cells illuminated with a red-light source of comparable power maintained high GFP expression, suggesting the photoswitchable response was blue light-dependent and not the result of sample heating or generic phototoxicity (Supplementary Fig. 3a). Shifting the insertion site by a single amino acid position to Lys23 produced a photoswitchable Gal4 with a reduced overall range and leakier gene expression under blue light (Fig. 3a), supporting prior studies in which the precise position of LOV domain insertion can strongly influence photoswitchable activity[5,20], even for neighboring residues within the same loop. Finally, we also transfected the Gal4LOV^SK22-VP64 variant into two additional cell lines—mouse NIH3T3 fibroblasts and human SUM159 breast cancer cells—in which the UAS-GFP construct was stably integrated, obtaining at least a 100-fold difference in GFP expression between light and dark in each case (Supplementary Fig. 3b, c).

We next sought to further characterize the LightsOut system by measuring its dose response, switching dynamics, and ability to produce tissue-scale gene expression patterns. We first constructed a stable 293-LPR cell line with Gal4LOV^SK22-VP64 inserted at the landing pad locus (which we termed 293-LightsOut cells); these cells exhibited similar photoswitchable responses to those previously obtained after the final library sort and in transient transfection (Fig. 3b). To perform a dose response, we exposed 293-LightsOut cells to light pulses at various duty cycles, from 2.5 s of light every 30 s of light to continuous illumination (Fig. 3c; Supplementary Fig. 4). We found that the LightsOut system exhibits a smooth, tunable response between light dose and GFP expression. To measure the switching kinetics, we incubated 293-LightsOut cells in continuous dark or light for 24 h, and then switched to the inverse condition and monitored destabilized GFP expression over time (Fig. 3d). We observed a 50% change in GFP levels within 6 h after the change in illumination conditions, with cells reaching GFP expression levels as high as cells kept in constant dark within 12 h after a shift to darkness, and to low GFP levels within ~20 h after the shift to blue light. These results compare favorably to the kinetics of previously published light-switchable Gal4 systems (LightOn half-life ~30 h; ShineGal4 half-life ~5 h)[18,21], and are generally consistent with rapid light-induced changes in gene expression as our assay is limited by the time required for transcription, translation, chromophore maturation, and

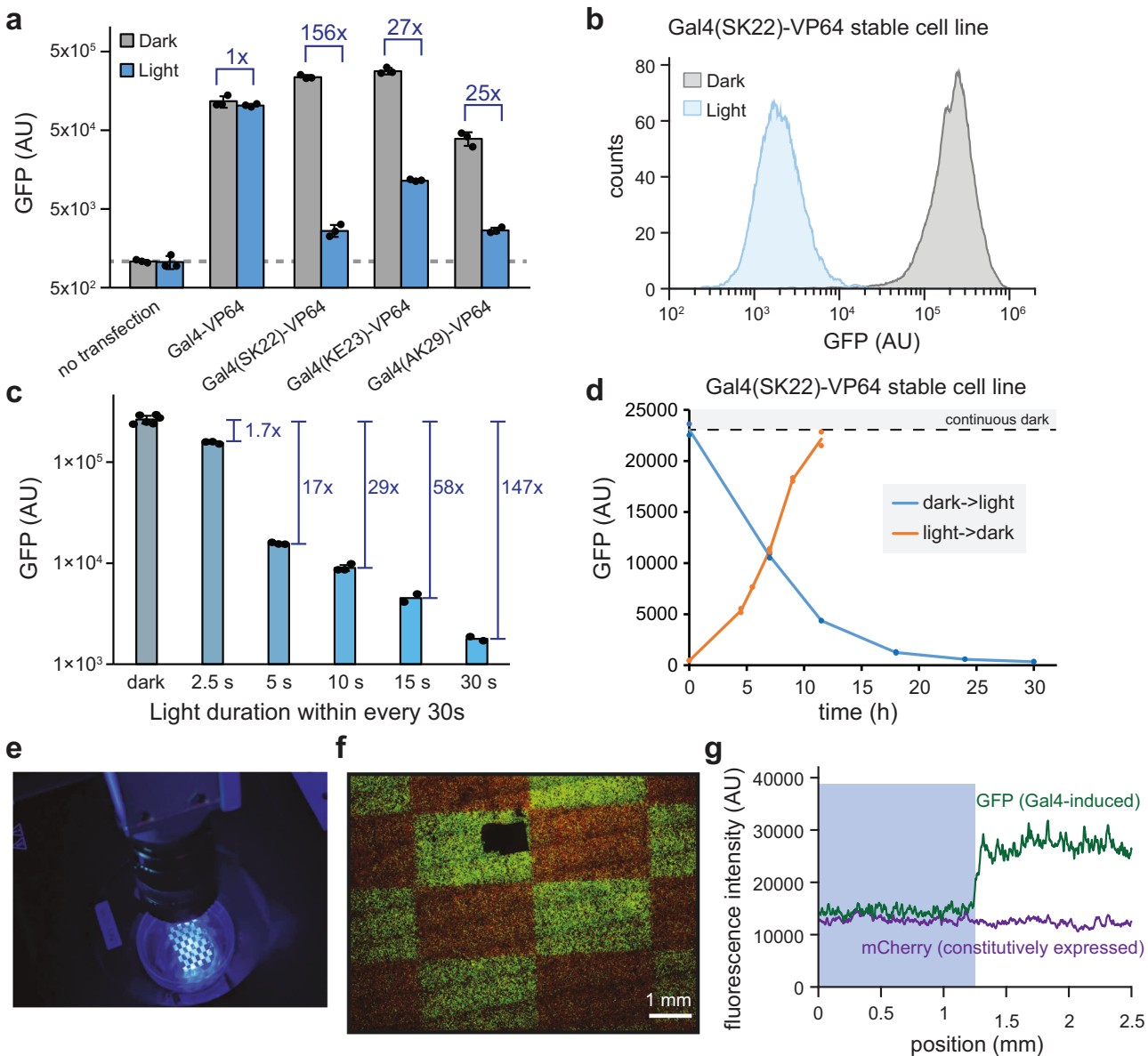

**Fig. 3 | Characterization of the LightsOut transcription factor. a** Light-dependent GFP expression of 293-LPR cells transfected with unmodified Gal4-VP64 or variants harboring AsLOV2 insertions at the indicated sites. Cells were incubated in dark or light for 22 h before GFP measurement by flow cytometry. Three biological replicates were measured for each condition, and the error bars indicate mean ± SEM. **b** Light-dependent response of 293-LPR cells harboring the best-performing Gal4LOV^SK22-VP64 LightsOut variant. **c** GFP expression of 293-LPR cells harboring Gal4LOV^SK22-VP64 and treated with different pulse sequences of light, where the autofluorescence of plain HEK293T cells was subtracted before fold-change calculation. Six biological replicates for dark condition, two biological replicates for 15 s and constant light conditions, and three biological replicates for

other conditions were measured. Error bars (only when n > 2) indicate mean ± SEM. **d** Determination of reversibility and kinetics of the LightsOut transcription factor by keeping cells in dark or light for 24 h, then switching to the inverse condition and measuring GFP expression by flow cytometry. **e, f** Spatial illumination drives patterning of LightsOut-expressing cells. By projecting a spatial pattern onto a monolayer of cells (**e**), we obtained a checkerboard pattern of GFP expression (**f**). All cells constitutively express mCherry. Similar checkerboard pattern was obtained in two independent repeat experiments. **g** Quantification of GFP intensity across a spatial illumination boundary from cells in (**f**), revealing tight spatial control for cm-scale overall patterns. Source data for (**a**–**d**) and (**g**) are provided as a Source Data file. AU arbitrary units.

accumulation of destabilized GFP (which has a ~ 2 h half-life)[22]. Taken together, these data demonstrate that photoswitchable transcription by Gal4LOV^SK22-VP64 can be tuned to intermediate levels or reversibly switched between high and low states.

The high dynamic range of light-dependent transcription suggested that the LightsOut system could be used to deliver high-resolution spatial patterns of gene expression at a tissue scale. To investigate this possibility, we plated 293-LPR LightsOut cells in a confluent monolayer on 35 mm dishes and projected blue light patterns onto the monolayer using a digital micromirror device (Fig. 3e; see Methods). A checkerboard pattern of alternating darkness and

blue light illumination produced a clear spatial GFP pattern on the HEK293T cell monolayer (Fig. 3f). Quantification revealed that patterns could be formed with a spatial length scale of ~100 μm (Fig. 3g) even when delivering cm-scale overall patterns, demonstrating that the LightsOut system is suitable for delivering patterns of gene expression at tissue scales.

## Dissecting the processes underlying light-switchable Gal4LOV^SK22 function

What is the mechanism of the light-dependent change in Gal4-VP64 activity obtained by AsLOV2 insertion? The insertion between Ser22

and Lys23 is only one amino acid away from a conserved cysteine that coordinates a zinc ion as part of the conserved $Zn_2Cys_6$ zinc finger motif (Fig. 4a). We mapped the insertion onto a previously determined crystal structure of the Gal4 DNA binding domain in complex with DNA (PDB: 1D66)[23], revealing that this insertion site is far from the DNA-protein binding interface (Fig. 4b). This observation suggested that the light-dependent change in AsLOV2 conformation may act to disrupt the conformation of the zinc finger motif, either by allosterically affecting DNA binding or by changing the Gal4-AsLOV2 chimera's conformation in some manner that alters its transcriptional efficacy. However, many alternative potential mechanisms are also possible, including but not limited to: (1). Sequestration or inhibition of the VP64 transactivation domain by lit-state AsLOV2, (2) oxidative damage to either the cysteine-zinc coordination center or the DNA itself by flavin photoexcitation[24], (3) misfolding and degradation of the Gal4-AsLOV2 transcription factor after light stimulation, or (4) a light-induced change in transcription factor localization (e.g., nuclear export)[5,25].

We first replaced the VP64 with a commonly-used transactivation domain from the p65/RelA transcription factor[18] to test whether our results depended on the specific transactivation sequence used. We constructed a Gal4LOV[SK22]-RelA fusion construct as well as a control variant lacking AsLOV2 insertion, then tested their transcriptional activity in light and dark in HEK293T UAS-dGFP cell lines (Supplementary Fig. 5). Just as previously observed for Gal4LOV[SK22]-VP64, cells expressing Gal4LOV[SK22]-RelA exhibited a strong photoswitchable response, demonstrating that light-controlled gene expression does not depend on the specific transactivation sequence used.

We next investigated how light-dependent gene expression varied as a function of the linker between AsLOV2 and the Gal4 DNA binding domain (Fig. 4c). Our rationale was that longer, more flexible linkers would dramatically decrease conformational coupling between these domains and weaken light responses, whereas non-allosteric mechanisms such as oxidative damage by flavin photoexcitation would be unaffected[26,27]. We transfected each variant into 293-LPR cells and measured GFP after overnight incubation in dark or blue light (Fig. 4d). For the shortest construct—the short AsLOV2[408–543] variant lacking any linker residues—we observed low GFP expression in both dark and light, suggesting a pre-distorted Gal4 conformation that was unable to drive gene expression (Fig. 4d, left). At the other extreme, the largest variant (the canonical AsLOV2[404–546] domain with six linker residues) produced high gene expression in both light and dark, consistent with weak conformational coupling to the Gal4 domain. Similar results were obtained for a second series of variants with finer resolution of linker lengths ranging from 0-13 amino acids (Supplementary Fig. 6).

To further rule out the oxidative damage model, we next constructed a fusion between AsLOV2 and the N-terminus of Gal4-VP64, which is very close to DNA-protein binding interface (Fig. 4a), reasoning that this variant might retain localized oxidative damage but lack conformational coupling. The resulting fusion protein exhibited no light sensitivity at all (Supplementary Fig. 7). Taken together, these data argue against light-induced oxidative damage and support a model where conformational coupling between AsLOV2 and Gal4 drives the photoswitchable response, as has been reported in other contexts[2–4]. Furthermore, this data underscores the utility of the short AsLOV2[408–543] sequence in the context of transposase-based library construction. Transposition inevitably produces linkers between the inserted domain and target protein, and the short-LOV2 sequence appears to minimize any deleterious flexibility that this approach could produce.

We next set out to directly test whether light stimulation might distort the conformation of the Gal4 DNA binding domain and directly trigger unbinding from DNA. We performed fluorescence polarization between a Rhodamine Red-conjugated 27-bp DNA sequence

containing a 1xUAS element and a purified Gal4-AsLOV2[SK22] construct identical to our typical LightsOut protein but lacking the VP64 transcriptional activation domain (see Methods). We first performed spectroscopy on the purified protein to confirm the characteristic absorbance of AsLOV2's FMN cofactor at ~450 nm (Supplementary Fig. 8). We then titrated a 2 nM solution of UAS-containing DNA with increasing amounts of Gal4-AsLOV2[SK22], measured fluorescence anisotropy after light or dark exposure, and fit the resulting data to estimate a dissociation constant $K_D$ of $6.4 \pm 1.1$ nM (Fig. 5b), comparable to prior estimates of Gal4 binding to a 1xUAS-containing DNA sequence (e.g., $13 \pm 4$ nM)[28–30]. Similar results were obtained in both dark and light, but we reasoned that this may reflect a ~2 min delay between illumination and measurement in our fluorimeter during which time the transcription factor might cycle back to the dark state and re-bind DNA, thereby masking the true change in binding affinity (Fig. 5a).

To further test whether light induces a change in DNA binding affinity, we sought to repeat these in vitro binding measurements using mutants that either (1) lock the LightsOut system in the light or dark state or (2) slow down the AsLOV2 photocycle. We tested four mutations, numbered by their position in the full-length Phototropin 1 (Fig. 5c): (1) C450V, a commonly used dark-state mimic that lacks the cysteine required for adduct formation with the FMN cofactor[31]; (2) I532E/A536E, which destabilizes Jα helix docking and partially mimics the lit state[32]; (3) Q513L, a mutant that exhibits a slow photocycle and has been reported to adopt a dark-state conformation[33]; and (4) V416L, a mutant with a very slow (>1 h) photocycle[34]. We first tested the transcriptional activity of each variant in the dark and light using our standard 293-LPR transfection assay (Fig. 5d). We found that the C450V mutant exhibited a weaker light-induced response and I532E/A536E mutant exhibited lower activity in the dark, but both still exhibited ~20-fold changes in gene expression, consistent with recent reports that even LOV domains harboring dark- and lit-mimetic mutations may be capable of phototransduction and light-induced conformational changes[35,36]. We also tested a second dark-state mimic, the Q513L mutant, which failed to exhibit a light-dependent change in gene expression, suggesting that this variant performs better as a true dark-state mimic in this context[33]. Importantly, we still observed a very low level of gene expression after illumination from the slowest-photocycling V416L mutant, providing a tractable context for obtaining lit-state DNA binding data by fluorescence anisotropy.

We performed fluorescence anisotropy measurements on Gal4-AsLOV2 variants harboring all four sets of mutations (Fig. 5e, f). Across all contexts, we broadly found a slight but clear trend between GFP expression in cells and DNA binding affinity. Dark-incubated Gal4-AsLOV2[SK22], the C450V mutant, and the I532E&A536E mutant showed progressively decreasing DNA binding affinity, from 6.4-13.4 nM (Fig. 5e), in line with the lower gene expression observed over this series (Fig. 5d). We also observed a two-fold change in DNA binding affinity for the long-lived V416L mutant between light and dark conditions (8.6 nM vs 16.6 nM). However, it is difficult to reconcile the ~200-fold change in gene expression with the relatively minor change in DNA binding affinity, particularly given affinities in the ~10 nM range even under conditions when no transcription was observed (e.g., V416L under blue light). Taken together, our data supports a model where light-induced conformational changes to the Gal4-AsLOV2 partially alter DNA binding, but it is likely that additional molecular processes are required to produce the potent transcriptional effects observed in cells.

Finally, we tested whether photoactivation of Gal4-AsLOV2[SK22] might lead to degradation and/or nuclear export of the fusion protein by transfecting 293-LPR cells with a fluorescent Gal4-AsLOV2[SK22]-VP64-mCherry fusion construct. We cultured cells in either sustained blue light or darkness for 22 h and measured mCherry (to report on the total quantity of the photoswitchable transcription factor or its localization in cells) and GFP (to assess transcriptional activity) in each cell

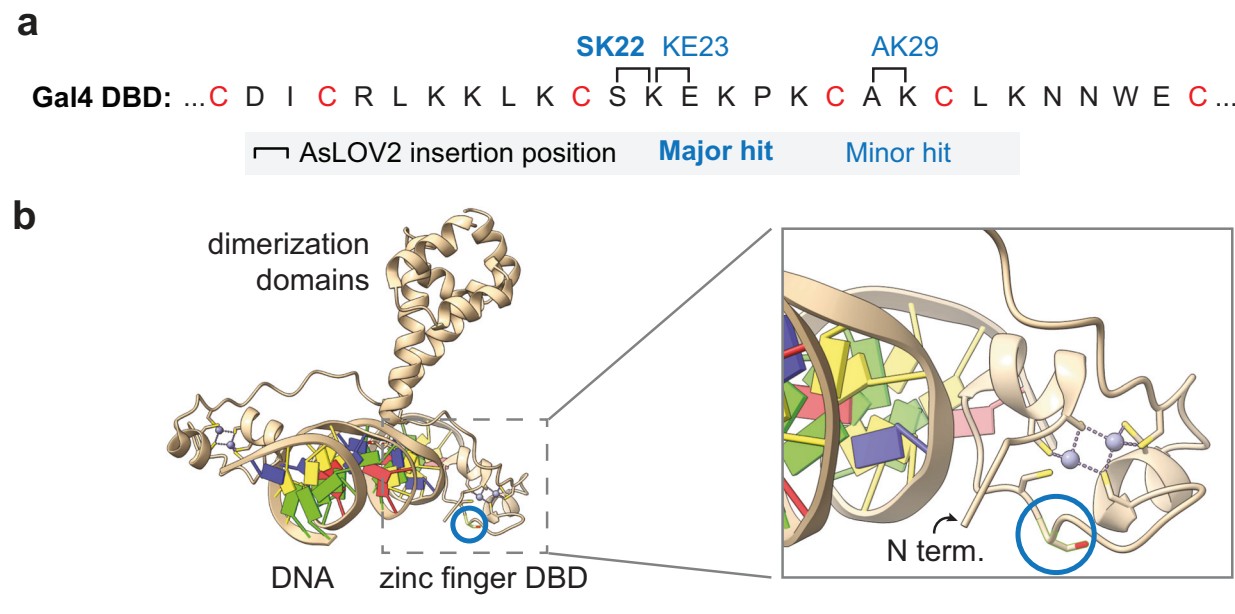

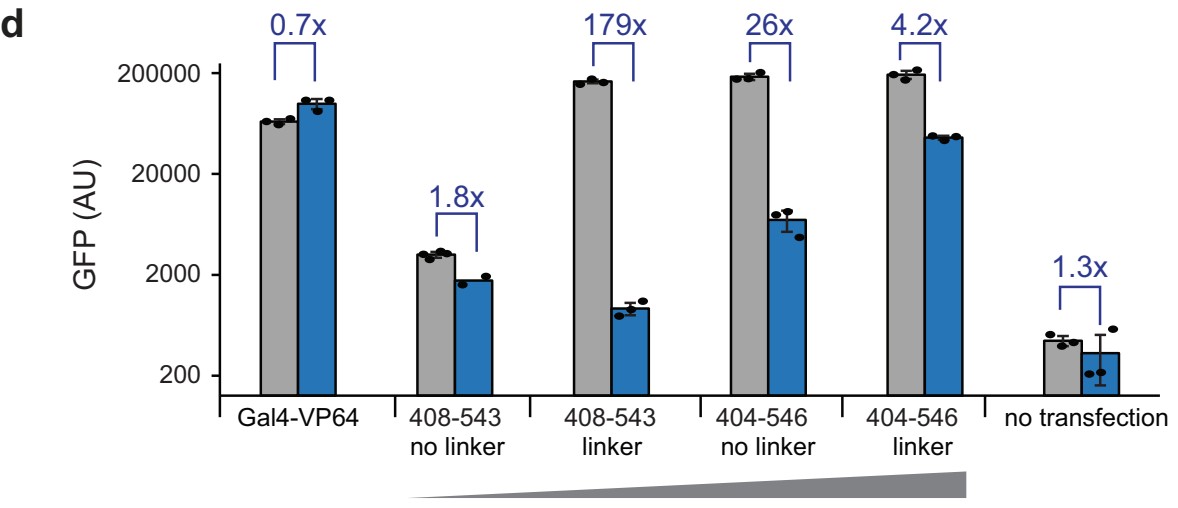

**Fig. 4 | Dissecting mechanisms of light-dependent transcriptional activity.**
**a** The amino acid sequence of Gal4 DNA binding domain's zinc finger motif, with the six conserved cysteines that coordinate two zinc ions shown in red. Three insertion sites from high-throughput screening are shown. **b** Mapping the SK22 insertion onto a crystal structure of the Gal4 DNA binding domain. **c** Four insertion variants were designed to test how flexibility alters coupling between AsLOV2 and the Gal4 DNA binding domain. Short and long variants of AsLOV2 with linkers based on the Mu transposon scars were tested. **d** GFP expression of 293-LPR cells transfected with the four variants in (**c**) and incubated overnight in darkness or continuous blue light. Fluorescence values were background corrected by subtracting autofluorescence of plain 293 T cells. Four biological replicates for 408–543 no linker in dark condition, two biological replicates for 408–543 no linker in light condition, and three biological replicates for all other conditions were measured. Error bars (only when $n > 2$) indicate mean ± SEM. Source data for (**d**) are provided as a Source Data file. AU arbitrary units.

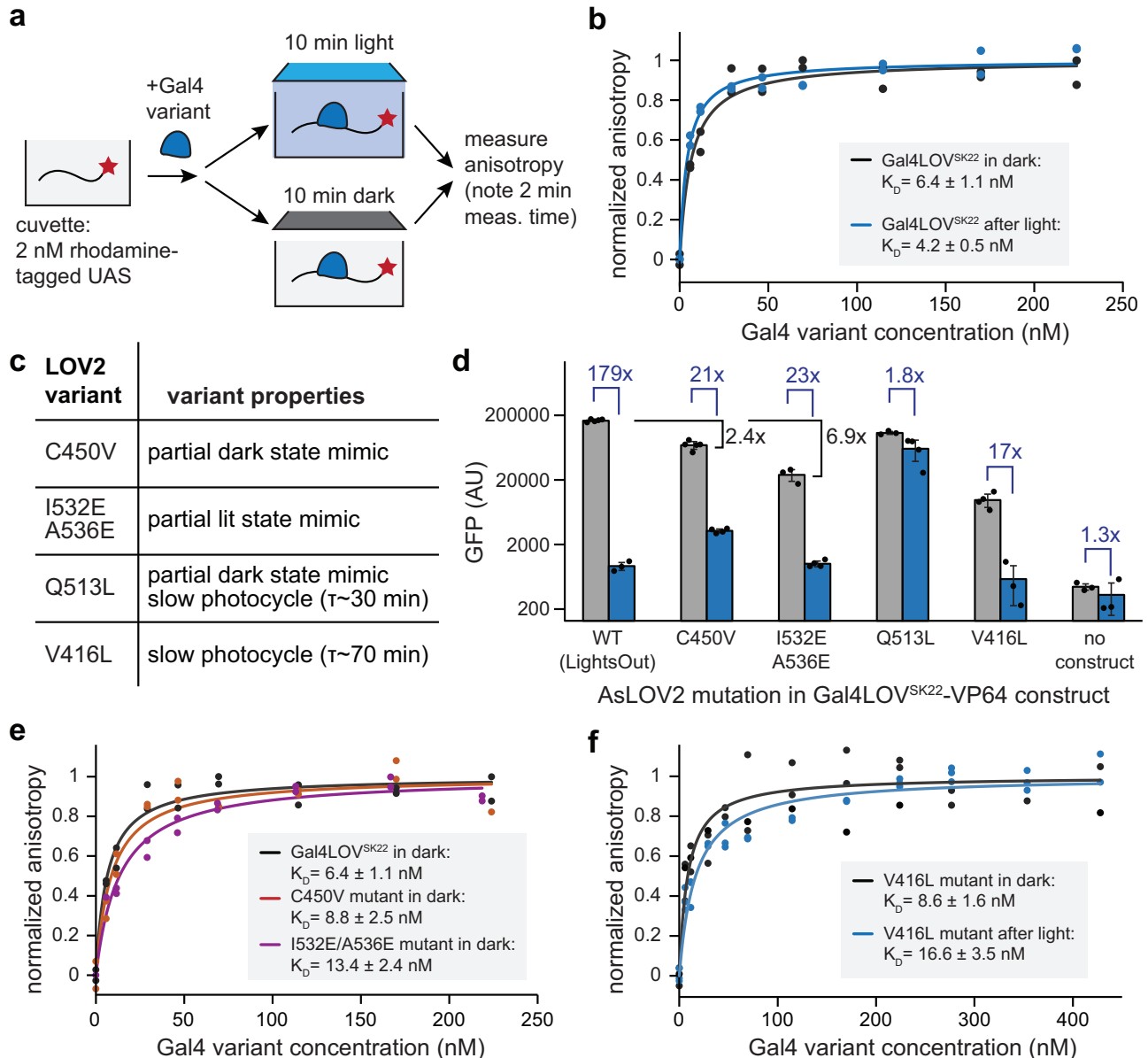

**Fig. 5 | Measuring changes in binding between Gal4LOV^SK22 and UAS-containing DNA in vitro. a** Schematic of fluorescence anisotropy measurements to determine the binding affinity of Gal4 and Gal4LOV(SK22) in light or dark to a 1×UAS DNA sequence in vitro. **b** Fluorescence anisotropy measurements for Gal4LOV^SK22 and 1xUAS DNA in light and dark. Two technical replicates were measured per condition, and data was fit to a one-site binding model (solid lines) to estimate the dissociation constants shown. **c** Table of AsLOV2 variants used for protein binding measurements, including candidate dark- and lit-state mimics as well as slow photocycling variants. **d** GFP expression of 293-LPR cells transfected with the mutants in (**c**) and incubated overnight in dark or blue light. Fluorescence values were background corrected by subtracting autofluorescence of plain 293 T cells. The fold-change between wild-type and C450V as well as between wild-type I532E/A536E in dark conditions are also indicated. Five biological replicates for WT and C450V in dark, four biological replicates for V416L in dark, four biological replicates for C450V, Q513L and I532E&A536E in light, and three biological replicates for other conditions were measured. The error bars indicate mean ± SEM. **e, f** Fluorescence anisotropy measurements for Gal4LOV^SK22 variants in (**c**). At least two technical replicates were measured per condition, and data was fit to a one-site binding model (solid lines) to estimate the dissociation constants shown. Source data for (**b, d–f**) are provided as a Source Data file.

by flow cytometry (Fig. 6a). We found that mCherry levels were similar for cells incubated in blue light and dark conditions despite profound differences in GFP expression, suggesting that the Gal4LOV^SK22-mCherry protein is not substantially degraded over at least 22 h in response to blue light illumination.

We also imaged these cells in light and dark conditions by confocal microscopy (Fig. 6b), which revealed some light-dependent changes in the subcellular distribution of the mCherry-tagged Gal4 construct. In the dark, Gal4LOVSK22-VP64-mCherry was distributed throughout the nucleus and cytosol, but in many light-stimulated cells we also observed the formation of a single peri-nuclear mCherry

cluster. However, quantification of nuclear fluorescence in single cells revealed that nuclear mCherry levels were overlapping between light and dark conditions, despite stark differences in GFP expression (Fig. 6c). To further test whether protein redistribution is sufficient for a change in GFP expression, we also constructed a fluorescent Gal4-AsLOV2^404–546 variant based on our prior observations that this construct retains high GFP expression in the light (Fig. 4d). Imaging this variant also revealed similar peri-nuclear clusters despite high GFP expression in light and dark (Fig. 6d), indicating that this light-induced redistribution of Gal4-AsLOV2 protein is not sufficient for the switch in transcriptional activity.

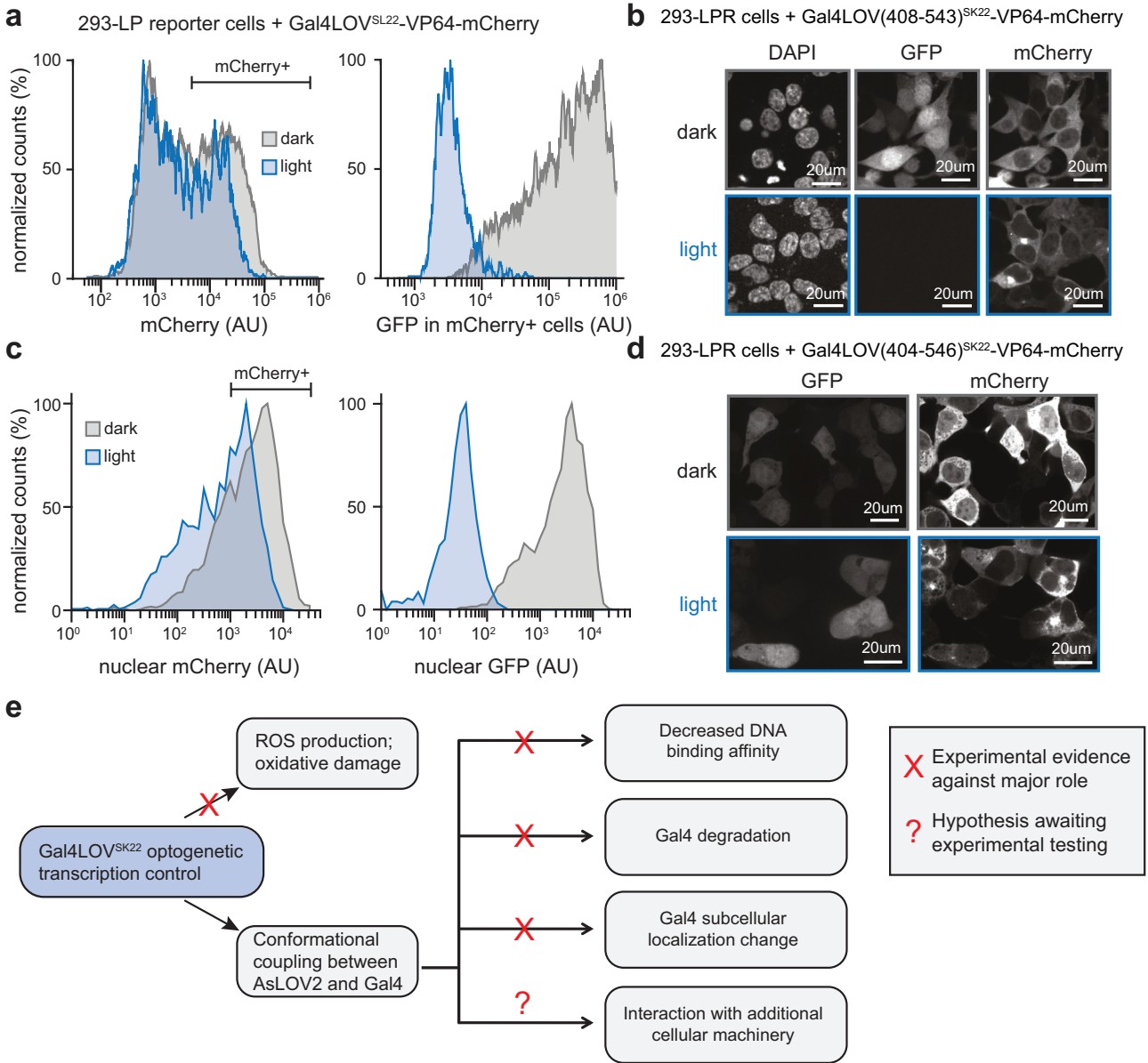

**Fig. 6 | Testing alternative mechanisms of light-dependent gene expression.**
**a** mCherry and GFP fluorescence of 293-LPR cells transfected with Gal4LOV^SK22^-VP64-mCherry, following 22 h incubation in continuous darkness or blue light.
**b** Fluorescence microscopy of 293-LPR cells transfected with Gal4LOV^SK22^-VP64-mCherry, following 22 h incubation in continuous darkness or blue light. Similar imaging results were obtained from two independent samples per condition.
**c** Quantification of single-cell nuclear mCherry (left) and GFP (right) from images in

(**b**) for at least 1700 cells per condition. GFP levels are for mCherry+ cells, gated as shown. **d** Fluorescence microscopy of 293-LPR cells transfected with Gal4LOV(404–546)^SK22^-VP64-mCherry, fixed, and imaged after 22 h in dark or blue light conditions. Similar imaging results were obtained from two independent samples per condition. **e** Schematic of mechanistic studies for explaining light-induced changes in gene expression. Source data for (**a** and **c**) are provided as a Source Data file. AU arbitrary units.

In sum, our data indicates that light-dependent transcriptional switching is dependent on conformational coupling between Gal4 and AsLOV2 at the SK22 insertion site, yet no mechanism that we tested appears to fully account for the large transcriptional differences we observe (Fig. 6e). It is possible that light-switchable transcription is the product of multiple mechanisms acting in concert (e.g., the combined action of a change in DNA binding affinity, protein abundance, and protein redistribution). It is also possible that additional mechanisms are at play beyond what has been tested here, such as an interaction with other cellular machinery (e.g., chaperone proteins) that might selectively bind to the distorted conformation of Gal4-AsLOV2^SK22^ produced by blue light and prevent DNA binding or transcription in cells. More broadly, these results underscore the value of an unbiased screening approach in mammalian cells for discovering novel light-switchable protein variants, which can in principle produce potent light-switchable activity through a wide range of detailed molecular processes.

## Extending light-switchable function to other zinc-finger DNA binding domains

Since we obtained several hits of AsLOV2 insertion into Gal4's $Zn_2Cys_6$ zinc finger DNA binding domain, we hypothesized that similar insertion sites might produce photoswitchable gene expression in other related transcription factors. We first turned to the *Neurospora crassa* Qf protein, a second $Zn_2Cys_6$ zinc finger transcription factor that shares a high degree of similarity to Gal4, particularly in the DNA binding domain, and which binds to a well-characterized QUAS DNA sequence[37]. We chose sites for short LOV2 (408–543) insertion near the

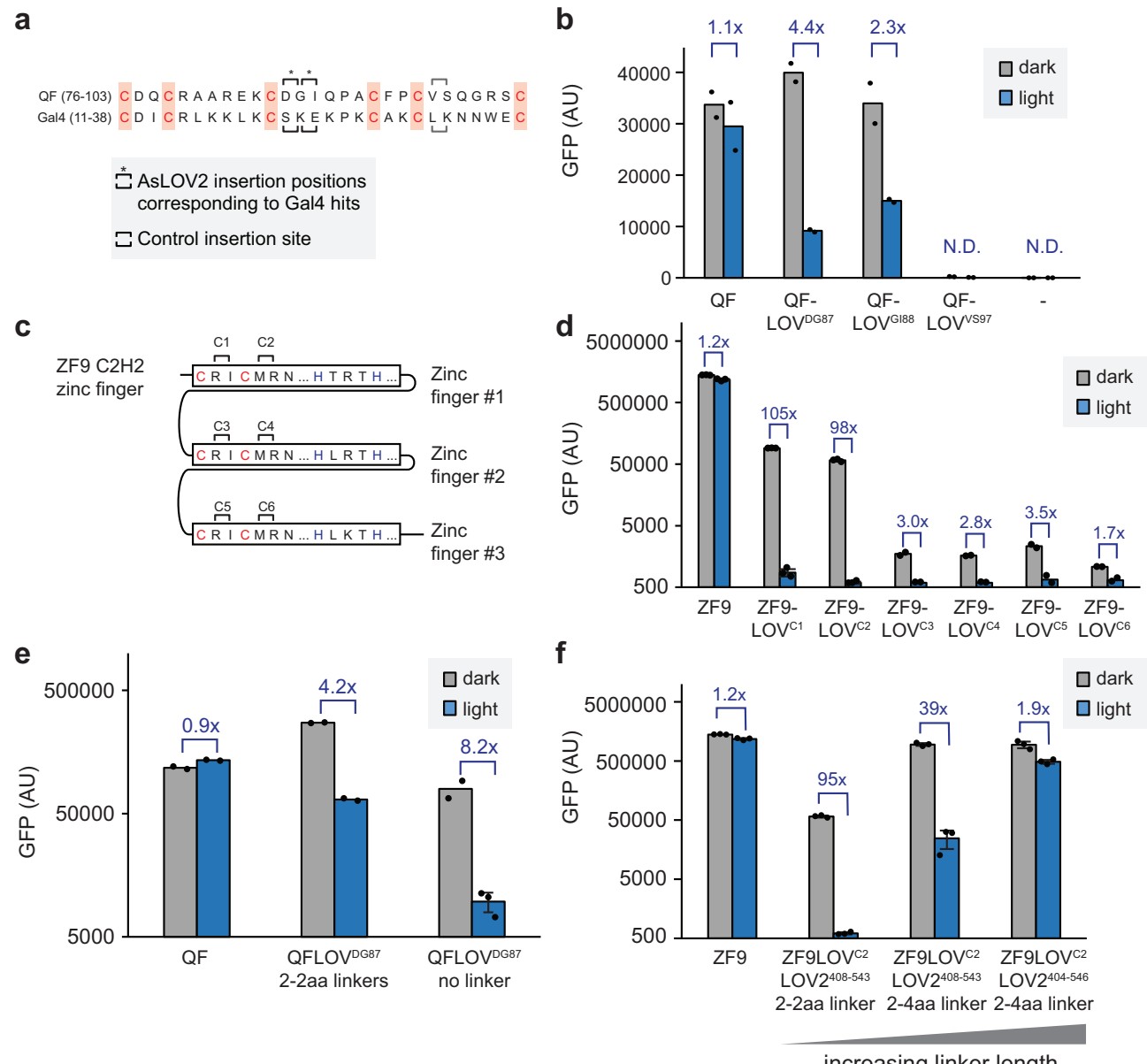

**Fig. 7 | Generalizing photoswitchable gene expression to additional zinc finger transcription factors. a** Alignments of the $Zn_2Cys_6$ zinc finger motifs from the Qf and Gal4 proteins, with six conserved cysteines coordinated to two zinc ions shown in red. Insertion sites tested for Qf are marked by brackets, with asterisks marking previously identified insertion sites from the Gal4 screen, and gray brackets indicate a similar insertion site following a cysteine that was not previously identified as a hit. **b** Testing for light-dependent transcriptional activity in Qf-AsLOV2 variants. HEK293T cells were transfected with the indicated Qf-AsLOV2 variants as well as QUAS-GFP plasmids, and GFP expression was measured by flow cytometry. Two biological replicates were measured for each condition, and the error bars indicate mean ± SEM. **c** The amino acid sequence of the tridactyl ZF9 synthetic DNA binding domain. Zinc-coordinating cysteine and histidine residues are shown in red and blue, respectively. The four AsLOV2 insertion sites tested are marked by brackets and labeled C1-C4. **d** Testing for light-dependent transcriptional activity in ZF9-

AsLOV2 variants. HEK293T cells were transfected with the indicated ZF9-AsLOV2 expression plasmid as well as a reporter plasmid harboring a ZF9-specific promoter driving GFP. Three biological replicates for ZF9, ZF9-LOV$^{C1}$ and ZF9-LOV$^{C2}$ conditions both in light and dark, two biological replicates for the rest conditions were measured Error bars (only when $n = 2$) indicate mean ± SEM. **e** GFP expression in 293 cells transfected with QUAS-dGFP and Qf-LOV$^{DG87}$ variants with different linkers between AsLOV2$^{408-543}$ and Qf. Three biological replicates for QFLOV$^{DG87}$ no linker in light, two biological replicates for all the rest conditions were measured, Error bar (where $n > 2$) indicate mean ± SEM. **f** GFP expression in 293 cells transfected with a ZF9-specific promoter driving GFP and ZF9-AsLOV2$^{C2}$ variants with different linkers between AsLOV2 and ZF9. Three replicates were measured for each condition and error bars indicate mean ± SEM. Source data for (**b**, **d–f**) are provided as a Source Data file. AU arbitrary units.

3rd zinc-conjugating cysteine (QfLOV$^{DG87}$-VP64 and QfLOV$^{GI88}$-VP64) flanked by 2 amino acid linkers on each side, analogous to two of the hits from our Gal4 screen (Fig. 7a). We transfected 293 T cells with a QUAS-dGFP reporter plasmid and individual Qf variants and measured GFP levels by flow cytometry after incubating cells in blue light or darkness for 22 h (Fig. 7b). Meanwhile, we also tested an additional control insertion site at an analogous position following the 5th Cys

residue that was not found in the initial screen to test if AsLOV2 insertion might generally alter DNA binding activity near any Cys residue (Fig. 7a). We found that AsLOV2 insertion indeed produced some photoswitchable gene expression in the Qf transcription factor, albeit with a substantially reduced range (4.4-fold change in gene expression, vs 156-fold for Gal4). The relative ordering of activity for each insertion site was also preserved, with a weaker light-induced

transcription observed for the QfLOV[GI88] variant due to a higher baseline of expression in the light, just as had been previously noted for the corresponding Gal4LOV[KE23] variant (Fig. 4a). Finally, we found that AsLOV2 insertion after the 5th Cys residue produced a non-functional transcription factor, with no GFP accumulation in either light or dark, consistent with its failure to be detected as a hit in our initial screen.

We next tested whether the AsLOV2 insertion strategy might generalize even beyond the $Zn_2Cys_6$ class of zinc fingers when applied near Cys residues in other zinc finger family DNA binding domains, such as the C2H2 family of zinc fingers that plays key roles in transcription and genome organization in higher eukaryotes[38]. As a model system we chose ZF9[39], a synthetic zinc finger protein composed of three C2H2 domains with six structural cysteines in total (2 per C2H2 domain). We tested all six cysteine-proximal insertion sites, transfecting 293 T cells with plasmids harboring each ZF9-AsLOV2 insertion variant (flanked by 2 amino acid linkers on each side) and a ZF9-responsive promoter driving GFP and performing flow cytometry after incubation under blue light or darkness for 22 h. We observed a photoswitchable response when AsLOV2 was inserted at either of two positions in the first C2H2 cluster of ZF9 (Fig. 7d). Both of these variants exhibited nearly 100-fold reduction in GFP expression upon illumination, albeit with reduced transcriptional activity in the dark compared to unmodified ZF9. We observed some photoswitchable transcriptional but overall poor activity for AsLOV2 insertion in the second or third C2H2 cluster. Taken together, the results obtained in this section confirm that the major result of our screen—photoswitchable gene expression driven by AsLOV2 insertion near structural Cys residues in zinc finger DNA binding domains—can be generalized across multiple transcription factors.

Based on our data that varying the linkers between LOV2 and Gal4 had substantial effects on photoswitchable transcription, we next tested if varying linkers could be used to optimize the performance of AsLOV2-inserted QF and ZF9. For QF-LOV[DG87], we observed high GFP expression even under blue light (Fig. 7b), suggesting inefficient conformational coupling between AsLOV2 and QF. Indeed, removing the 2 amino acid linkers on both sides of AsLOV2 produced a greater photoswitchable GFP response (8.2 vs 4.2-fold), albeit at the expense of lower dark-state gene expression (Fig. 7e). For ZF9-LOV[C2], we observed reduced GFP expression in the dark compared to wild type ZF9 (Fig. 7d), suggesting a pre-distorted conformation. Indeed, slightly increasing the linker length resulted in higher GFP expression in both dark and light while maintaining a 39-fold difference between these states, whereas an even longer variant resulted in complete loss of photoswitchable activity (Fig. 7f). Taken together, these results demonstrate that slight variations of linker sequences can be used to tune optogenetic transcription factor performance, enabling rapid optimization from an initial insertion site.

## Discussion

Over the past decade the toolbox of cellular optogenetics has dramatically expanded, and a variety of optimized light-sensitive effectors are available to control protein-protein association[40–43], clustering[40,44–47], nuclear-cytosolic transport[25,48,49], kinase activity[2,50,51], and gene expression[17,18,21]. Nevertheless, the development of new optogenetic tools is typically still a bespoke process, requiring the replacement of a protein's natural mode of regulation (e.g., eliminating Gal4's natural ability to dimerize) with a photoswitchable analog (e.g., fusion of a light-inducible dimerization domain to the Gal4 DNA binding domain)[18]. Both steps require detailed information about a protein's structure and function, and consequently light-sensitive variants are only available for a limited number of well-studied targets. We reasoned that unbiased screening offers a path toward obtaining new light-gated biological functions, as it can be applied to a broad range of targets and requires minimal prior knowledge about their

structure or function. Previous foundational studies have performed bacterial screens for light-regulated dihydroxy folate reductase (DHFR) and Cas9 function using targeted AsLOV2 insertion at 70 or 234 surface residues, respectively, obtaining protein variants with 2–7-fold changes in light-dependent activity[3,8]. However, unbiased screens have not yet been applied in mammalian cells, perhaps due to challenges inherent in constructing a library of light-sensitive variants at comparable expression levels and carrying out successive selections under light and dark conditions.

Here, we report a pipeline for generating and screening libraries of candidate optogenetic tools in mammalian cells. Using the Gal4-VP64 transcription factor as a model system, we establish libraries in which AsLOV2 is inserted at >80% of possible nucleotide positions in any of the three possible reading frames. By inserting the library into a defined locus in mammalian cells, we were then able to directly screen for photoswitchable function by incubating cells in light or dark conditions and sorting for desired GFP levels in each case. Our screen identified a Gal4 variant with strongly photoswitchable responses, achieving a >150-fold increase in gene expression between light and dark conditions. Moreover, we find that the site of AsLOV2 insertion generalizes to other zinc finger transcription factors, including a member of the C2H2 family, the largest class of human transcription factors and one for which few photoswitchable variants currently exist[49]. Because the size of domain insertion libraries scales with the number of amino acids in the target protein, this approach can retain high coverage even for large target proteins or when multiple variants of the inserted domain are used (e.g., with varying linker sequences) while maintaining high overall coverage.

Our approach relies on a key feature: the ability to couple photoswitchable protein function to a readout that is compatible with cell sorting. This coupling is easiest to achieve when the photoswitchable protein is itself a transcription factor, motivating our choice of the potent Gal4-VP64 transcription factor as an initial target. However, we note that many intracellular processes can be linked, directly or indirectly, to the expression of a fluorescent protein: chromatin modification using synthetic reporter cassettes[52], protein-protein interactions using split transcription factors[53], and cell signaling using pathway-specific enhancers and promoters[54,55]. In principle, our approach could thus enable discovery of light-gated effectors in each case. Moreover, we note that the AsLOV2 insertion approach has already proven its versatility, producing photoswitchable nanobodies, monobodies, transcription factors, and kinases, with some exhibiting high lit-state activity others having high dark-state activity depending on the specific insertion site and target protein[5]. Thus, by modifying the sorting criteria at each stage of selection, it is likely possible to identify both light-activated and light-inactivated protein variants, even within a single library.

Our screen identified a single optimal AsLOV2 insertion site in Gal4, following Ser 22 in its Zn2Cys6 DNA binding domain. A series of variants harboring different mutations or linker residues supports a model where conformational coupling AsLOV2 and Gal4 drives the change in transcriptional response. However, we were surprised to find that the potent and rapid light-dependent switch in Gal4-VP64 transcriptional activity cannot be easily ascribed to a similarly large change in protein expression, nuclear localization, or in vitro DNA binding affinity. It is possible that the change in LightsOut transcriptional activity is a product of a combination of slight changes in each of these parameters, or that additional cellular processes not measured here (e.g., light-dependent chaperone binding or transcriptional machinery recruitment) are also regulated by our insertion variant. Nevertheless, our study underscores the power of employing a phenotypic screening approach directly in mammalian cells[56,57]: a rational design strategy that only focused on generating a photoswitchable DNA binding domain would likely have failed to identify this AsLOV2 insertion site with exceptionally potent control over transcription.

While our initial screen focuses on the model Gal4-VP64 transcription factor, we provide some evidence that the zinc finger-targeted insertion sites we found may generalize beyond this single case. Analogous AsLOV2 insertions—at the +1 position near Zn-conjugating cysteine residues—generated light-dependent transcriptional responses for two additional transcription factors, in one case producing a C2H2-family transcription factor with a ~100-fold change in gene expression. We also establish a route toward optimizing dynamic range or maximum expression level by varying linker sequences. Further optimization might lead to additional improvements, either by testing additional linker sequences, using site-directed mutagenesis to improve conformational coupling between AsLOV2 and the DNA binding domain, or performing additional screens using these alternative targets. Along with foundational studies screening libraries using phage display and bacterial cells[3,8,43], our study suggests that systematic, high-throughput tools have much to offer in the discovery and refinement of the next generation of optogenetic tools.

## Methods

### Plasmid construction

Plasmids related to Mu transposition were originally obtained from Addgene which include pUCKanR-Mu-BsaI (Addgene #79769) and pATT-Dest (Addgene #79770). pLZA066 was constructed by cloning Gal4(1-147)-VP64 into pATT-Dest through in-fusion assembly (TaKaRa #638911) with BsmBI restriction sites on both side of Gal4-VP64. Landing pad plasmids including recombination plasmids pJG_082 AttB_mCherry_Bgl, pJG_083 AttB_EGFP_Bgl2 and thepJG_075 Bxb1 which expresses Integrase essential for gene integration are gifts from Jacob Goell (Rice University, Hilton Lab). pLZA063 was constructed by cloning Gal4(1-147)-VP64 into attB recombination plasmid through in-fusion assembly, and IRES-mCherry, the PCR product with pIRES2-mCherry-p53 deltaN (Addgene #49243) as template was tagged to Gal4-VP64 also through in-fusion assembly. Plasmids related to QF and ZF9 were originally obtained from Addgene including QF-encoding plasmids pCMV-QF (Addgene #24339), QF reporter plasmid pQUAS-luc2 (Addgene #24337), ZF9-encoding plasmid pVITRO1-SS-113 (Addgene #68737) and ZF9 reporter plasmid pGL4.26-SS-192 (Addgene #68759). In-fusion assembly was used to insert AsLOV2 into certain positions of Gal4-VP64, QF and ZF9 for light-switchable behavior testing. See Supplementary Table 2 for important plasmids used in this study.

### AsLOV2 insertion library

To get the transposon with encoded chloramphenicol resistance gene, PCR was performed with pUCKanR-Mu-BsaI as template, 5′-tagg-cacccaggctttacac-3′ as forward primer and 5′-tctgtaagcggatgccggga-3′ as reverse primer. The product was purified and digested with HindIII and BglII followed by purification with NucleoSpin Gel and PCR Clean-up Columns (Takara, #740609). The resulting DNA was directly used as transposon. Transposition reactions were conducted in a total volume of 20 μL with the following components: 0.45 pmol transposon DNA, 0.13 pmol pLZA066, 4 μL 5×MuA reaction buffer and 1 μL 0.22 μg/μL MuA transposase enzyme (Thermo Fisher Scientific, #F750). Reactions were incubated for 18 h at 30 °C followed by 10 min at 75 °C to heat inactivate MuA transposase. Completed reactions were cleaned up with DNA Clean & Concentrator-5 Kit (Zymo Research, #D4014) and eluted with 20 μL nuclease-free water (Thermo Fisher Scientific, #AM9932), which was then transformed into 200 μL TransforMax Electrocompetent E. coli (Lucigen, #EC10010). An aliquot of the recovery culture was spread on an LB agar plate with carbenicillin (Gold Biotechnology, #C-103-5) and chloramphenicol (Gold Biotechnology, #C-105-5) antibiotics to assess reaction efficiency. Remaining recovery culture was transferred to 50 ml LB with chloramphenicol and carbenicillin to select for plasmids with transposon

insertion, and after overnight growth the library was collected with the Plasmid Plus Midi Kit (Qiagen, #12943), which was named pLZA066_CmR01.

Golden Gate cloning was used to replace the chloramphenicol resistance gene with the AsLOV2 domain. The library purified from the last step was directly used as the backbone and insert was linear DNA with AsLOV2 (408−543) coding sequence and two BsaI digestion sites on both ends. To accommodate all possible insertions with adequate reading frames, three kinds of insert were used and named as LOV01 (with one additional base in the 5′ end of AsLOV2), LOV02 (with two additional bases both in the 5′ and 3′ end of AsLOV2) and LOV03 (with one additional base in the 3′ end of AsLOV2). Briefly, 46fmol of backbone and 88fmol of AsLOV2 insert was mixed with 15 units BsaI_HFv2 (New England Biolabs, #R3733S), 800 units T4 DNA Ligase (New England Biolabs, #M0202S) and 1×T4 DNA Ligase Reaction Buffer in a total volume of 20 μL. The reaction was incubated 2 min at 37 °C, 5 min at 16 °C (first two steps cycled 50 times), 20 min at 60 °C and 20 min at 80 °C. Reactions were purified with DNA Clean & Concentrator-5 Kit and transformed into 50 μL TransforMax Electrocompetent E. coli. Cells were then transferred to 25 ml LB with carbenicillin, and the library was collected with the Plasmid Plus Midi Kit, which was named pLZA066_LOV01, pLZA066_LOV02, pLZA066_LOV03 (corresponding to the insert LOV01, LOV02 and LOV03).

To isolate the Gal4-VP64 fragments with AsLOV2 insertions, the library was then digested with BsmBI_v2 (NEB, #R0739S) and AsLOV2 inserted Gal4-VP64 was gel purified from the mixture. Golden Gate cloning was used again to clone these Gal4-VP64 library into landing pad recombination plasmid. The gel purified products were used as inserts and the BsmBI-digested, linearized landing pad recombination plasmid was used as the backbone. Briefly, 39 fmol backbone and 78 fmol of insert were mixed with 15 units BsmBI_v2, 800 units T4 DNA Ligase and 1×T4 DNA Ligase Reaction Buffer in a total volume of 20 μL. The reaction was incubated 2 min at 42 °C, 5 min at 16 °C (first two steps cycled 50 times), 20 min at 60 °C and 20 min at 80 °C. Reactions were purified using a DNA Clean & Concentrator-5 Kit and transformed into 50 μL TransforMax Electrocompetent E. coli. Cells were then transferred to 25 ml LB with carbenicillin, and the library was collected with the Plasmid Plus Midi Kit, which were named pLZA063_LOV01, pLZA063_LOV02 and pLZA063_LOV03. See Supplementary Table 3 for all libraries constructed for this study.

### Lentivirus transduction and clonal cell line sorting

293 T landing pad (293T-LP) cell line was a gift from Kenneth Matreyek (Case Western Reserve University), which was derived from 293 T that had been bought from Thermo Fisher Scientific. 293 T landing pad reporter cell line (293T-LPR) was constructed by incorporating UAS-dGFP into 293T-LP genome through lentivirus transduction, with iRFP as the marker. We chose to use lentivirus because it is a rapid approach for generating stable cell lines. Although the lentiviral approach produced random integration that can lead to cell-to-cell differences in expression, we selected a single clonal cell line harboring a single integration site for all subsequent comparisons using our library and individual Gal4-AsLOV2 variants. All the cells were kept in DMEM (ThermoFisherScientific, #11995073) supplemented with 10% FBS (R&D Systems, #S11150), 1% Pen Strip (Gibco, #15140-122) and 2mM L-Glutamine (Gibco, #25030-081) throughout all experiments.

To produce lentiviral particles, HEK 293 T cells were plated on a 6-well plate grown up to 40% confluency. At that point they were co-transfected with pLZA042 and lentiviral packaging plasmids (pMD and CMV) with FuGENE HD (Promega, #E2311). Specifically, 1500 ng pLZA042, 1330 ng pCMVdR8.91 and 170 ng pMD2.G mixed with 9 μL Fugene HD transfection reagent were used for each well in the 6-well plate. Virus was collected after approximately 48 h, filtered using a 0.45 mm filter. Polybrene (Sigma-Aldrich, #TR-1003-G) was added to the viral particles to the final concentration of 5 μg/mL. 293 T LP cell

line was plated on a 6-well plate and infected with 200–500 µL of the virus at 40% confluency, and iRFP positive clonal cell was sorted at least 48 h post the infection time. Two weeks after clonal sorting several clones were characterized through flow cytometry and a few clones showing uniform iRFP expression were chosen as 293 T LP UAS-dGFP cell line (293 T LPR) for following screening experiments. See Supplementary Table 4 for all stable cell lines used in this study.

## Integration of Gal4 AsLOV2 insertion library into landing pad cell line

293 T LPR cell line were plated on 6-well plates one day before transfection. Recombination was performed by transfecting cells with 1500 ng of pJG75 Bxb1 and 2500 ng of AsLOV2 insertion library pLZA063_LOV01, pLZA063_LOV02 or pLZA063_LOV03 (for each individual well) in doxycycline-free media with FuGENE HD transfection reagent. Two or more days following transfection, the media was changed to media supplemented with 2 µg/mL doxycycline. Three days after media replacement cells that are mCherry positive and BFP negative were sorted to a new 6-well plate with flow cytometry.

## Library screening

Library cells were kept in dark when culturing. One day before screening, cell media was changed to doxycycline media and 4 h post media replacement cells were irradiated with 450 nm blue LEDs at an intensity of 0.34 mW/cm² or kept in dark for 22 h. When ready for characterization and sorting, cells were detached with trypsin, and resuspended in DMEM containing 10% serum. Flow cytometry was performed with SH800S Cell Sorter equipped with Sony 100 µm Sorting Chip. EGFP was excited with a 488 nm laser, and emitted light was collected after passing through 525/50 nm band pass filters. mCherry was excited with a 561 nm laser, and emitted light was collected after passing through 600/60 nm band pass filters. Before sorting, live, single cells were gated using FSC-A and SSC-A (for live cells) and FSC-A and FSC-H (for single cells) and at least 100,000 cells were sorted and plated on a new 6-well plate. See Supplementary Table 5 for all sorted cell libraries used in this study.

## Extraction of genomes from cells and next-generation sequencing

PureLink Genomic DNA Mini kit (Invitrogen, #K182001) was used to extract genomes from library cells following the protocol in manual, and PCR was performed with forward primer 5′-CCAGGGCTCGAGAC CGCAACTACACGCCACC-3′ and reverse primer 5′-AGCTTCGAATTCG GGGCGGATCAGCTTGGTAC-3′ to amplify the library fragments from the genome. Library fragment DNA was then sheared with a Covaris S220 focused ultrasonicator using AFA microTUBEs (Covaris, #PN 520052) to an approximate size of 300–400 base pairs. NEBNext® Ultra™ II DNA Library Prep with Sample Purification Beads (New England Biolabs, #E7103S) was used to prepare DNA library for sequencing from sheared DNA. Sheared DNA and prepared samples were analyzed for size distribution on an Agilent 2100 Bioanalyzer using DNA 1000 chips (Agilent Technologies). Double-stranded DNA concentrations of the adaptor-prepared samples were measured with a dsDNA HS Assay Kit (Invitrogen, #Q32851) on a Qubit Fluorometer. A normalized pool of samples was run on a MiSeq Nano 300nt or MiSeq Micro 300nt for 318 cycles. Analysis of FASTQ files of sequencing results was performed through MATLAB R2021b, with the script provided in Supplementary Note 4.

## Cell transient transfection and blue light irradiation

293 T LP cells or 293 T LPR cell line were plated on 12-well plates 1 day prior to transfection. 800 ng of DNA was used when plasmids encoding Gal4 variants were transfected to 293 T LPR (for each individual well) while 1000 ng DNA (in total) was used when both QF or ZF9-encoding plasmid and corresponding reporter plasmid (each 500 ng)

were transfected to 293 T LP cells (for each individual well). The FuGENE HD Transfection Reagent was used for all the transfections. All the cells were kept in the dark for 5 h post transfection and then irradiated with 450 nm blue LEDs at an intensity of 0.34 mW/cm² or remained in the dark for additional 22 h before characterization. Final GFP levels were characterized by flow cytometry. For analysis, single cells were gated using FSC-A and SSC-A (for live cells) and FSC-A and FSC-H (for single cells). See Supplementary Fig. 9 for gating strategy. Transfection experiments were also performed with NIH3T3 cells and SUM0159 cells with the same protocol. SUM159s were obtained as a gift from Dr. Yibin Kang (Princeton Univ.). NIH3T3 cells were received as a gift from Dr. Wendell Lim (UCSF) and authenticated by STR profiling (ATCC).

## Protein expression and purification

Gal4LOVSK22 or its mutants were expressed in E. coli strain BL21(DE3) (Sigma-Aldrich, # CMC0016) at 18 °C for 20 h in the presence of 0.5 mM IPTG (GoldBio, #I2481C25), 10 µM ZnCl2 (Sigma-Aldrich, #39059-100ML-F) and 5 µM FMN (Sigma-Aldrich, #F2253-25MG). The cell pellet was collected by centrifugation and sonicated in buffer A containing 20 mM Tris (Quality Biological, #351-006-721), 0.5 M NaCl (Sigma-Aldrich, #59223), 10 µM ZnCl2, 20 mM imidazole (CEPHAM Life Sciences, #10371), 10 mM β-mercaptoethanol (Sigma-Aldrich, #M6250) and 10% glycerol (Sigma-Aldrich, #G7893), pH 7.5. The soluble cell lysate was fractionated by centrifugation. The supernatant was passed over a Ni-NTA column (QIAGEN, #1018142) then washed thoroughly with buffer A, and finally eluted with buffer B containing 20 mM Tris, 0.5 M NaCl, 50 µM ZnCl2, 500 mM imidazole, 10 mM β-mercaptoethanol and 10% glycerol, pH 7.5.Buffer exchange was then performed with ultracentrifuge and the storage buffer contains 20 mM HEPES (Thermo Scientific, #J60712.AK), 0.15 M NaCl, 20 µM ZnCl2, and 10% glycerol, pH 7.5. Proteins were then aliquoted and stored in −80 after purification.

## Fluorescence anisotropy experiments

The probes used were as follows: /5RhoR-XN/-TCTTCGGAGGGCTG TCACCCGAATATA-3′ (IDT) and its complementary strand (does not contain the fluorophore). The DNA was annealed with 1 µM each strand in water. Fluorescence anisotropy was conducted with the Fluorolog from Horiba with temperature control and fluorescence anisotropy modules in a semi-micro quartz cuvette with light path 10 × 4 mm (Hellma. #114F-10-40). For measurement, DNA was diluted to 2 nM in 800 µL renaturation buffer containing 20 mM HEPES (pH = 7.5), 50 mM NaCl, 50 µg/mL BSA (EMD Millipore, #2960-500GM) and 5% Ficoll 400 (Sigma-Aldrich, #F2637-5G), then titrated with protein starting at 6.2 nM until the anisotropy values are saturated. For binding measurement in dark, the sample was kept in dark for 10 min before anisotropy measurement at each titration. For binding measurement in blue light, the sample was irradiated with 450 nm blue LEDs at an intensity of 1.9 mW/cm² for 10 min before anisotropy measurement at each titration.

## Spatial illumination

Spatial patterns of light were delivered using a DLP 4500 Lightcrafter module (Texas Instruments) comprising a digital micromirror device with an integrated 450 nm LED. Optical patterns were re-imaged onto the sample plane using a macrophotography lens (Carl Zeiss 100 mm f/2 Makro-Planar T* 2/100) using a custom-built setup. Patterns were applied with a mean intensity of 0.9 mW/cm² for 24 h, during which cell culture conditions (37 °C, 5% CO2, >95% relative humidity) were maintained using an environmental chamber (Tokaii Hit INUG2AH-TIZSH).

## Cell imaging

For imaging, 35 mm glass bottom dishes with 20 mm well (Cellvis, #D35-20-1.5-N) were used. Glass was first treated with 10 µg/mL of fibronectin in PBS for 30 min in 37oC. 293 T UAS-dGFP cell line was plated on dish

and allowed to adhere onto the plate. For mCherry tagged Gal4LOVSK22 tracking experiment, 2500 ng of pLZA144 encoding Gal4LOVSK22-VP64-mCherry was transfected to 293 T UAS-dGFP cells with the FuGENE HD Transfection Reagent following the protocol in manual and kept in dark first. 4 h after transfection, cells were either switched to blue light illumination or kept in dark for another 20 h. Then cell media was aspirated and cells was fixed in 4% PFA in PBS for 15 min in room temperature, followed by PBS washing for three times. Cells were then stained with 2 µg/mL DAPI for 15 min in room temperature, followed by PBS wash for three times and kept in 4oC before imaging. Imaging was done using Nikon Eclipse Ti microscope with a Prior linear motorized stage, a Yokogawa CSU-X1 spinning disk, an Agilent laser line module containing 405, 488, 561 and 650 nm lasers, an iXon DU897 EMCCD camera, and a 40X oil immersion objective lens. Several images of fixed cells in the 405, 488 and 561 channels were collected.

### Statistics & reproducibility
No statistical method was used to predetermine sample size. The light/dark selection and sorting of our AsLOV2 insertion library was performed only once since it is very time-consuming. All other experiments were repeated at least twice with similar results, and the number of replicates for each condition or experiment are stated in main text and supplementary information. All attempts at replication were successful. No data were excluded from the analyses. Different conditions were randomly assigned to the wells on a 12-well plate for randomization. The Investigators were not blinded to allocation during experiments and outcome assessment.

### Reporting summary
Further information on research design is available in the Nature Portfolio Reporting Summary linked to this article.

## Data availability
There are no restrictions on availability. All raw data generated in this study are provided in the Supplementary Information/Source Data file. The original FASTQ files of all DNA sequencing results have been deposited in the NIH Sequence Read Archive under the accession code PRJNA974403. All plasmids and cell lines presented in this study will be shared upon request from the corresponding author. Source data are provided with this paper.

## Code availability
The code for analyzing next-generation sequencing data is provided in Supplementary Note 4.

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

## Acknowledgements

We thank all members of the Toettcher lab helpful comments. This work was supported by NIH grant U01DK127429, an Eric and Wendy Schmidt Transformative Technology Award, and the Princeton Catalysis Initiative (J.E.T.). We highly appreciate Sarah Gernhart and Prof. Haw Yang for allowing us to perform fluorescence polarization assay with their fluorometer. We also thank Jacob Goell and Prof. Isaac Hilton in Rice University for providing the landing pad plasmids as well as Prof. Kenneth Matreyek (Case Western Reserve University) for the 293 T landing pad cell line, and we thank Prof. Alexander Ploss and Prof. Nieng Yan for allowing the use of centrifuges in their labs.

## Author contributions

Conceptualization, L.Z. and J.E.T.; Investigation, L.Z., H.M.; Writing, L.Z. and J.E.T.; Funding Acquisition, J.E.T.; Supervision, J.E.T.

## Competing interests

J.E.T. is a scientific advisor for Prolific Machines and Nereid Therapeutics. H.M.M. is a cofounder and scientific advisor for C16 Biosciences. The remaining authors declare no competing interests.
