## [Peer Review File · Nature Communications]

Reviewers' Comments:

Reviewer #1:

Remarks to the Author:

I reviewed a previous version of this manuscript. I consider it a very strong contribution to the field of optogenetic protein engineering. The authors have added significant new data that strengthen the manuscript. However, two features could be improved:

The authors tested the functionality of 4 variants with different linker lengths (no linker, LightsOut, full length LOV with/without linker). In the introduction they argue that even one residue change to a linker can alter allosteric coupling. These four variants have the linker length (both linkers) of: 0, 6, 13, and 19. This is rather an extreme jump in linker length. The fact that LightsOut exhibits the best performance is not very surprising since it was perhaps optimized for that position during the selection. While this result argues for allosteric coupling and a lack of effect of ROS, it is not strong evidence that the Mu scar is the best linker length.

Because, this is a methodology paper, protein engineers will wish to know how much linker optimization is likely necessary. In the Savage paper, the linker composition was very important, perhaps because cationic residues near the insertion site can alter the ionization state of the GFP chromophore. People who want to use this method with LOV insertions will wonder whether they also will need to do another round of linker deletion/optimization to obtain better photoswitchable tools (admittedly 150-fold is pretty good in this case, but it is one example).

The lack of a mechanism remains problematic. Have the authors considered the light state LOV may somehow sequester/block VP64? Perhaps some modelling may suggest if this is plausible (<https://pubs.acs.org/doi/10.1021/bi0482912>). If this AD is swapped for another one (p65? As in <https://www.ncbi.nlm.nih.gov/pmc/articles/PMC7491154/#mmc1>) does the system still work?

Reviewer #2:

Remarks to the Author:

Zhu et al present a compelling manuscript where they outline a new methodology to allow for the construction of a complete library of LOV domain insertions into a protein of choice. Specifically, they leverage the recent developments of the "landing pad" system to insert AsLOV2 into a Gal4-VP64 system. The strategy is robust, and allows one to screen every possible insertion point under conditions that will yield equivalent expression levels, and allow for an in cell readout of function that can easily be assayed in a high throughput manner. As a result, they were able to identify unique insertion sites that enable light to turn-off expression of a GFP reporter. These insertion sites flank key cysteine residues, and they are able to prove that the ability to regulate expression with light is not limited to Gal4, but rather insertion near similar Cys residues in other Zinc finger transcription factors. Overall, the method is compelling and affords a new tool in the construction of optogenetic tools.

The manuscript is well written, easy to read, and should be accessible to a wide audience, and is of significance to diverse fields.

Below I outline only minor critiques.

Minor Comments:

1. The overall screening method is very robust and although the resulting insertion site is not optimal for all ZF proteins (e.g. QF system) the insertion strategy does clearly extend to other ZF proteins, including the engineered ZF9 system and then likely many others. This is a major advancement for the field.

2. Line 292: Typo, should read "may act to disrupt" or "may disrupt" not "may act disrupt".

Reviewer #3:

Remarks to the Author:

As the reviewer has commented in the first round of the peer review, methodologies used for screening and development of the presented blue light-inducible transcription factors, such as library construction based on Mu transposase and cell-based library screening, has been commonly used in the field of protein engineering and are not novel. The authors also describe in Introduction (line 97-97) that "Our strategy relies on the Mu transposase to randomly insert the LOV sequence into a target protein of interest (ref#11), followed by genomic integration of individual library members into a mammalian cell line using the recently developed "landing pad" system (ref#12,13)". However, throughout the manuscript, the authors describe as if their methodologies/approaches are novel. For example, the authors describe in Abstract (line 37-38) that "Here, we describe a strategy for generating and screening a library for novel optogenetic tools directly in mammalian cells". Since these descriptions may be misleading to the readers, the reviewer suggests the authors to revise the relevant explanations throughout the manuscript.

The reviewer confirmed that the authors have responded to other three comments raised in the first round of the peer review.

Response to reviewers

Reviewer comments shown in black; our responses are shown in blue.

Referee #1 (Remarks to the Author):

I reviewed a previous version of this manuscript. I consider it a very strong contribution to the field of optogenetic protein engineering. The authors have added significant new data that strengthen the manuscript. However, two features could be improved:

The authors tested the functionality of 4 variants with different linker lengths (no linker, LightsOut, full length LOV with/without linker). In the introduction they argue that even one residue change to a linker can alter allosteric coupling. These four variants have the linker length (both linkers) of: 0, 6, 13, and 19. This is rather an extreme jump in linker length. The fact that LightsOut exhibits the best performance is not very surprising since it was perhaps optimized for that position during the selection.

While this result argues for allosteric coupling and a lack of effect of ROS, it is not strong evidence that the Mu scar is the best linker length.

Because, this is a methodology paper, protein engineers will wish to know how much linker optimization is likely necessary. In the Savage paper, the linker composition was very important, perhaps because cationic residues near the insertion site can alter the ionization state of the GFP chromophore. People who want to use this method with LOV insertions will wonder whether they also will need to do another round of linker deletion/optimization to obtain better photoswitchable tools (admittedly 150-fold is pretty good in this case, but it is one example).

We appreciate the comment & agree that a finer sampling of linker lengths would be useful and appropriate. We now test nine different Gal4LOVSK22 variants with a variety of linker lengths ranging from 0-13 amino acids (**Figure S6**). Our engineered LightsOut variant, whose total linker length is 6 amino acids, corresponds to the middle of this range.

Overall, these new data reveal a clear trend of light-inducible responses. The variants with linker length between 4 amino acids to 7 amino acids all exhibited similarly strong photoswitchable responses. In contrast, we observed a much weaker light response as the total number of linker residues was increased from 7 to 9 amino acids, and very short linkers (0-2 aa) produced very low transcriptional activity even in the dark, presumably due to a distorted Gal4 DNA binding domain.

In sum, it appears that a potent photoswitchable response can be achieved across a range of linker lengths. In general, we would still recommend linker optimization when engineering any new optogenetic protein variant, particularly because the response likely also depends on the flexibility of the loop in which the LOV domain is being inserted.

The lack of a mechanism remains problematic. Have the authors considered the light state LOV may somehow sequester/block VP64? Perhaps some modelling may suggest if this is plausible

(<https://pubs.acs.org/doi/10.1021/bi0482912>). If this AD is swapped for another one (p65? As in <https://www.ncbi.nlm.nih.gov/pmc/articles/PMC7491154/#mmc1>) does the system still work?

We thank the reviewer for providing another excellent plausible hypothesis. In the newly revised manuscript, we tried swapping the VP64 for another transactivation domain, that of p65/RelA, but still obtained a similar light-switchable transcriptional response.

Overall, we have strived to test many possible mechanisms for light-induced gene expression and hope that the reviewer agrees that our current study now merits publication. We are already planning follow-up experiments as we seek to extend our approach to a much broader class of photoswitchable transcription factors and look forward to reporting any new mechanistic insights that we obtain in future studies.

Referee #2 (Remarks to the Author):

Zhu et al present a compelling manuscript where they outline a new methodology to allow for the construction of a complete library of LOV domain insertions into a protein of choice. Specifically, they leverage the recent developments of the “landing pad” system to insert AsLOV2 into a Gal4-VP64 system. The strategy is robust, and allows one to screen every possible insertion point under conditions that will yield equivalent expression levels, and allow for an in cell readout of function that can easily be assayed in a high throughput manner. As a result, they were able to identify unique insertion sites that enable light to turn-off expression of a GFP reporter. These insertion sites flank key cysteine residues, and they are able to prove that the ability to regulate expression with light is not limited to Gal4, but rather insertion near similar Cys residues in other Zinc finger transcription factors. Overall, the method is compelling and affords a new tool in the construction of optogenetic tools.

The manuscript is well written, easy to read, and should be accessible to a wide audience, and is of significance to diverse fields.

Below I outline only minor critiques.

Minor Comments:

1. The overall screening method is very robust and although the resulting insertion site is not optimal for all ZF proteins (e.g. QF system) the insertion strategy does clearly extend to other ZF proteins, including the engineered ZF9 system and then likely many others. This is a major advancement for the field.

2. Line 292: Typo, should read “may act to disrupt” or “may disrupt” not “may act disrupt”.

We strongly appreciate the reviewer’s comment on our revised manuscript! We changed the typo in the text accordingly.

Referee #3 (Remarks to the Author):

As the reviewer has commented in the first round of the peer review, methodologies used for screening and development of the presented blue light-inducible transcription factors, such as library construction based on Mu transposase and cell-based library screening, has been commonly used in the field of protein engineering and are not novel. The authors also describe in Introduction (line 97-97) that “Our strategy relies on the Mu transposase to randomly insert the LOV sequence into a target protein of interest (ref#11), followed by genomic integration of individual library members into a mammalian cell line using the recently developed “landing pad” system (ref#12,13)”. However, throughout the manuscript, the authors describe as if their methodologies/approaches are novel. For example, the authors describe in Abstract (line 37-38) that “Here, we describe a strategy for generating and screening a library for novel optogenetic tools directly in mammalian cells”. Since these descriptions may be misleading to the readers, the reviewer suggests the authors to revise the relevant explanations throughout the manuscript.

The reviewer confirmed that the authors have responded to other three comments raised in the first round of the peer review.

We thank the reviewer for their thoughtful suggestions throughout the review process, and completely agree that the approach we describe combines multiple previously-developed protein engineering techniques, although to our knowledge they have not previously been applied in the context of optogenetic tool development. This use case did require some special considerations, such as sequential selection in light and dark conditions.

The reviewer raises a specific sentence that we think might represent a misunderstanding. We intended to describe a strategy to find novel optogenetic tools. We are not claiming the *strategies* of landing pad insertion or Mu transposition are themselves novel, but that the resulting optogenetic tools would be! As the Reviewer mentions, we have strived to cite the excellent prior studies that developed Mu transposition and the landing pad system throughout the paper. We have checked the entire manuscript and believe it is consistent in this regard.

Nevertheless, we are also happy to rewrite the sentence cited by the Reviewer to avoid any claims of novelty. It now reads: “Here, we adapt strategies for protein domain insertion and mammalian-cell expression to generate and screen a library of candidate optogenetic tools directly in mammalian cells.”

Reviewers' Comments:

Reviewer #1:

Remarks to the Author:

The authors have provided thorough and thoughtful responses backed up by new experiments. I fully support publication of this work.

Response to reviewers

We thank the reviewers for their time. No further comments were made by the reviewers that require a response.